# Diversity of Agro-Biological Traits and Development of Diseases in Alfalfa Cultivars during the Contrasting Vegetation Seasons

Aurelija Liatukienė *[ID], Eglė Norkevičienė, Vida Danytė and Žilvinas Liatukas

Institute of Agriculture, Lithuanian Research Centre for Agriculture and Forestry, Instituto al. 1,
LT-58344 Akademija, Lithuania
* Correspondence: aurelija.liatukiene@lammc.lt

**Abstract:** Alfalfa exhibits high adaptability to a range of environmental conditions. The aim of this study was to evaluate the agro-biological traits of alfalfa and select its most promising cultivars under different contrasting vegetation seasons. The field experiment was carried out at the Institute of Agriculture of Lithuanian Research Centre for Agriculture and Forestry. In 2016 and 2018, eleven alfalfa cultivars of different origin were established in the *Endocalcari Epigleyic Cambisol.* In the seasons of 2017–2018 and 2019–2020, the agro-biological traits of alfalfa cultivars were evaluated during winter; at the height of spring regrowth; before flowering at the three cuts; and during the fresh and dry matter yields; we also examined the development of diseases at the three cuts. The cultivar Birutė from Lithuania was distinguished by its wintering, its plant height at spring regrowth, its height before flowering, and its fresh and dry matter yields. The cultivar Timbale from France was distinguished by its wintering, spring regrowth, and its height before flowering. The cultivar Magnat from Romania was distinguished by its height before flowering and the fact that it was less damaged by downy mildew. The cultivars Jõgeva 118, Juurlu, and Karlu from Estonia were distinguished by their wintering and the fact that they were less damaged by diseases. The cultivar Eugenia from Italy was distinguished by the fresh and dry matter yields and the height before flowering.

**Keywords:** alfalfa; cultivar; disease; fresh and dry matter yield; plant height; wintering





## 1. Introduction

Alfafa (*Medicago sativa* L.) is one of the oldest and most important fodder plants and is widely distributed throughout different continents around the world [1,2]. Alfalfa is grown under a wide range of environmental and climatic conditions with variable precipitation and soil [3–6]. Alfalfa is a forage legume that demonstrates high forage yield potential, good persistence, and great tolerance to drought and frost. The extensive genetic and phenotypic variations found in alfalfa allow its cultivation in diverse climates and conditions [7].

The winter survival of alfalfa plants can be compromised by inadequate hardening conditions during the autumn and inadequate snow cover during the winter [8,9]. Precipitation and temperature fluctuations associated with climate change have an influence on plant winter survival [8]. Appropriate cultivation measures are also an effective way to improve cold resistance. Water plays an important role in the winter hardiness of alfalfa because freezing injuries are mainly caused by cell dehydration [10–12]. Water not only affects the cold resistance of alfalfa by changing the morphology and spatial distribution of the root system, but can also protect cells from low temperatures [13]. The root system is key to alfalfa's ability to resist low temperatures. Alfalfa plants can regulate root-system development in response to dynamic changes in soil moisture [14,15]. Kavut and Avcioglu [16] argued that soil factors such as nutritional, biological, or physiological factors, including soil temperature, influence root growth and development and the penetration of roots into the soil and thereby affect the yield of crops.

Alfalfa plants are damaged by fungal diseases, and one of these diseases is spring black stem leaf spot (*Phoma medicaginis* var. *medicaginis*). This disease spreads over the

overground parts of plants and limits the production of herbage and seed yield of alfalfa, causing considerable yield losses, especially seed yield, under wet weather conditions [17]. *Phoma medicaginis* not only reduces forage yield, but also reduces forage quality due to the influence of phytoestrogens adversely affecting the health of livestock [18,19]. Moreover, this disease damages the roots of alfalfa and is considered a pathogen of roots; it spreads along with other fungal diseases during the winter period [20]. Besides this, downy mildew, caused by *Peronospora trifoliorum* is also a devastating disease that affects alfalfa in temperate climate areas. Investigations on alfalfa's resistance to fungal diseases showed that material from different origins showed considerably different resistance to downy mildew [21]. Severe development of this disease reduces forage yields, especially seed yield and quality of crops [22,23].

Some cultivars and populations of alfalfa developed after several selection cycles using recurrent phenotypic selection of individual plants, which means that they have become more resistant to diseases, drought, and cold, and have a higher yield, as well as better nutritive value [4,24,25]. Therefore, to increase alfalfa planting areas in the country, it is important to find the alfalfa varieties suitable for different ecological regions and, most importantly, proper soil textures [26].

The aim of this study is to examine the agro-biological traits of alfalfa cultivars in the contrasting seasons, identify important traits, and select the most promising cultivars for breeding.

## 2. Materials and Methods

### 2.1. Experimental Material and Research Site

The experimental material consisted of 11 alfalfa cultivars of different geographic origin: the cultivar Birutė from Lithuania; the cultivars Karlu, Jõgeva 118, and Juurlu from Estonia; the cultivars Cosmina, Sigma, Magnat, and Alina from Romania; the cultivar Timbale from France; and the cultivars Gea and Eugenia from Italy. The experiment was carried out for growing seasons in 2017–2018 for the 2016 sowing year and in 2019–2020 for the 2018 sowing year at the research site of the Institute of Agriculture of Lithuanian Research Centre for Agriculture and Forestry, in the central part of Lithuania (55°23′ N lat., 23°51′ E long.).

The soil of the experimental site was *Endocalcari Epigleyic Cambisol* [27]. The physicochemical properties of soil investigated were: texture—light loam, pH (7.2–7.5), mobile $P_2O_5$ (201–270 mg kg$^{-1}$) and $K_2O$ (101–175 mg kg$^{-1}$), organic carbon $C_{org}$ (1.47%), and total nitrogen $N_{tot}$ (0.14–0.16%).

### 2.2. Treatments and Experimental Design

The cultivars of alfalfa were established after a black fallow without a cover crop within the first ten-day period of July in 2016 and 2018. A complex phosphorus (P) and potassium (K) fertilizer was applied once before sowing at a rate of $P_{60}K_{90}$. Each accession of alfalfa cultivars was sown in 5.5 m long and 1.5 m wide plots with four replications, with a small plot sowing machine, Hege. The field area of sowing of one alfalfa accession was 8.25 m$^2$. The experimental plots of alfalfa were sprayed with herbicide Basagran 480 (a.i. bentazon 480 g L$^{-1}$) 2 L ha$^{-1}$ to protect them from weeds in the 2016 and 2018 sowing years. After germination, alfalfa reached a height of 10 cm.

### 2.3. Evaluation of Agro-biological Traits in Experimental Fields of Alfalfa

The cultivars of alfalfa were evaluated during the seasons of 2017–2018 and 2019–2020. Wintering (W) was evaluated in early spring of each experimental year by (0–9 score scale); resistant (1–3 score); medium resistant–resistant (3–4 score); medium resistant (4–5 score); medium susceptible (5–6 score); medium susceptible–susceptible (6–7 score); susceptible (7 score); very susceptible–plant dead (8–9 score). The cultivars of alfalfa were evaluated for spring regrowth (SR). Two weeks after resumption of vegetation, the plant height was measured (cm). The plant height of alfalfa cultivars was measured before flowering (cm) at

the first, second, and third cuts (HF first, HF second, and HF third). The plant height in spring regrowth after resumption of vegetation and the height before flowering in each accession of alfalfa plots were measured for 30 plants in four replications. The plants of alfalfa cultivars were cut three times in each experimental year at an early flowering growth stage (10.0% flowers). The fresh matter yield (FMY) and dry matter yield (DMY) at the first, second and third cuts (FMY first, FMY second, and FMY third cuts; DMY first, DMY second, and DMY third cuts) were measured (t ha$^{-1}$). To determine dry grass content, fresh samples (500 g) of randomly chosen plants were taken from each plot, dried at 105 °C, and weighed. Downy mildew (DM) and spring black stem leaf spot (SBSLS) were evaluated in the seasons of 2017–2018 and 2019–2020. Disease severity was evaluated at the first cut (May–June), at the second cut (end July–August), and at the third cut (at the end of August–September) using the scale: 0.0, 0.1, 1.0, 5.0, 10.0, 20.0, 40.0, 60.0 and 80.0%. The fungal diseases were estimated by visually scoring each plot of alfalfa cultivars in four replications at the three cuts spring black stem leaf spot (SBSLS first, SBSLS second, SBSLS third cuts) and downy mildew (DM first, DM second, DM third cuts).

The area under the disease progress curve (AUDPC) of DM and SBSLS was calculated as the total area the graph of disease severity against time, from the first scoring to the last:

$$\text{AUDPC} = S_{I=1}^{n-1} \left[ (t_{i+1} - t_i)(y_i + y_{i+1})/2 \right] \tag{1}$$

where $t_i$—time (in day, hours, etc.) at the ith observation, $y_i$—an assessment of a disease (percentage, proportion, ordinal score, etc.) at the ith observation, and n—the total number of observations [28].

### 2.4. Statistical Analysis

Analysis of variance (ANOVA) was used, followed by Tukey's the least significant difference at the $p < 0.05$ significant levels. A one-way analysis of variance (ANOVA) was used to evaluate the date of wintering, the spring regrowth, the height before flowering in each cut, the fresh and dry matter yields in each cut, and resistance to diseases in each cut. The interactions between cultivars × year for W, SR, HF first, HF second, HF third cuts, FMY first, FMY second, FMY third cuts, DMY first, DMY second, DMY third cuts, SBSLS first, SBSLS second, SBSLS third cuts and DM first, DM second, and DM third cuts were evaluated using the two-factors analysis ANOVA. The relationships among the agro-biological traits were evaluated using correlations–regression analysis, between probability level ($p < 0.05$ and $p < 0.01$). All the data presented are the mean values of three independent sets of experiments (±standard error, SE). The statistical analyses were performed using the statistical program SAS Enterprise Guide, version 7.13 (SAS Institute Inc., Cary, NC, USA).

### 2.5. Meteorological Conditions during Experimental Year

The weather conditions during the winter period were wet and cool in the experimental years 2017–2020. In 2017, the weather conditions were rainy and warm during the vegetation period of alfalfa. The average temperature and precipitation were −2.4 °C and 19.7 mm in the winter season, 7.3 °C and 30.2 mm in the spring season, 16.5 °C and 93.0 mm in the summer season, and 8.3 °C and 84.5 mm in the autumn season. In 2018, the weather conditions were very warm and dry in the vegetation period. The average temperature and precipitation were −2.2 °C and 45.4 mm in the winter season, 8.3 °C and 36.5 mm in the spring season, 19.2 °C and 51.5 mm in the summer season, and 8.7 °C and 21.7 mm in the autumn season. In 2019, the weather conditions were dry and warm during the vegetation period. The average temperature and precipitation were −1.4 °C and 49.6 mm in the winter season, 8.4 °C and 30.8 mm in the spring season, 18.7 °C and 63.0 mm in the summer season, and 9.0 °C and 37.6 mm in the autumn season. In 2020, the weather conditions were very rainy and warm during the vegetation period. The average temperature and precipitation was 2.5 °C and 48.3 mm in the winter season, 7.0 °C and

30.4 mm in the spring season, 18.3 °C and 93.1 mm in the summer season, and 10.2 °C and 32.5 in the autumn season (Table 1).

**Table 1.** The amount of precipitation and air temperature during experimental period (data from the Dotnuva, Automatical Meteorological Station).

| Month | 2016 Sowing Year | | | | 2018 Sowing Year | | | |
|---|---|---|---|---|---|---|---|---|
| | 2017 | | 2018 | | 2019 | | 2020 | |
| | T, °C | P, mm | T, °C | P, mm | T, °C | P, mm | T, °C | P, mm |
| January | −3.2 | 14.2 | −1.6 | 55.5 | −4.4 | 48.8 | 2.7 | 52.2 |
| February | −1.6 | 25.1 | −6.1 | 16.2 | 1.2 | 39.2 | 2.3 | 47.3 |
| March | 3.5 | 38.9 | −1.9 | 17.3 | 3.3 | 37.0 | 3.5 | 31.7 |
| April | 5.6 | 48.2 | 9.9 | 52.1 | 8.9 | 0.0 | 6.8 | 9.4 |
| May | 12.8 | 3.4 | 16.9 | 40.2 | 12.9 | 55.4 | 10.6 | 50.1 |
| June | 15.4 | 72.1 | 17.5 | 34.1 | 20.6 | 16.1 | 18.9 | 165.9 |
| July | 16.7 | 153.8 | 20.5 | 83.3 | 17.3 | 66.0 | 17.4 | 65.6 |
| August | 17.3 | 53.2 | 19.5 | 37.0 | 18.2 | 107.0 | 18.5 | 47.7 |
| September | 13.3 | 123.1 | 15.0 | 19.1 | 12.8 | 48.5 | 15.0 | 14.8 |
| October | 7.4 | 89.3 | 8.3 | 32.8 | 9.3 | 34.9 | 10.2 | 49.4 |
| November | 4.2 | 41.0 | 2.9 | 13.1 | 4.9 | 29.5 | 5.3 | 33.3 |
| December | 1.0 | 64.4 | −1.1 | 60.8 | 2.4 | 45.3 | 0.6 | 24.4 |

Note. P = amount of precipitation, mm; T = mean temperature, T, °C.

## 3. Results

### 3.1. Wintering and the Height of Alfalfa Cultivars at the Spring Regrowth and before Flowering

The analysis of variance (ANOVA) in Table 2 revealed highly significant differences among the studied alfalfa cultivars for wintering, spring regrowth, and the height of plants before flowering at the first, second, and third cuts in both seasons. The results showed that the cultivars of alfalfa had variable responses to climatic conditions. The experimental year showed significant influence of wintering and the plant height at spring regrowth and before flowering during the first, second, third cuts in both experimental seasons. Additionally, year × cultivars interaction was significant for these traits (Table 2).

**Table 2.** The analysis of ANOVA and comparison of wintering, the height at spring regrowth, and at before flowering between seasons.

| Source | 2016 Sowing Year (2017–2018), *p*-Value | | | | | |
|---|---|---|---|---|---|---|
| | df | W | SR | HF 1st Cut | HF 2nd Cut | HF 3rd Cut |
| Cultivar | 10 | <0.0000 | <0.0000 | <0.0000 | <0.0000 | <0.0000 |
| Year | 1 | <0.0000 | <0.0000 | <0.0000 | <0.0000 | <0.0000 |
| Cultivar × Year | 10 | <0.0000 | <0.0000 | <0.0000 | <0.0000 | <0.0000 |
| | **2018 sowing year (2019–2020), *p*-value** | | | | | |
| Cultivar | 10 | <0.0000 | <0.0000 | <0.0000 | <0.0000 | <0.0000 |
| Year | 1 | <0.0000 | <0.0000 | <0.0000 | <0.0000 | <0.0000 |
| Cultivar × Year | 10 | <0.0000 | <0.0000 | <0.0000 | <0.0000 | <0.0000 |

| Trait | 2016 sowing year | | 2018 sowing year | |
|---|---|---|---|---|
| | 2017 | 2018 | 2019 | 2020 |
| W, Score | 2.0 a | 2.5 b | 2.6 a | 2.9 b |
| SR cm | 19.1 a | 29.1 b | 33.4 b | 20.3 a |
| HF 1st cut, cm | 95.5 b | 79.3 a | 98.8 b | 95.1 a |
| HF 2nd cut, cm | 75.2 b | 63.4 a | 57.9 a | 68.3 a |
| HF 3rd cut, cm | 63.6 b | 50.8 a | 62.0 a | 57.4 a |

Note. df = degree of freedom; W = wintering; SR = the height at spring regrowth; HF = the height before flowering. Different superscripts letters "a,b" in the same row in each year indicate difference in the level of *p* < 0.05.

In 2017, the cultivars Jõgeva 118, Juurlu, Timbale and Birutė were 1.2-fold better by wintering compared to cultivar Sigma (Figure 1). In 2018, the cultivars Jõgeva 118, Juurlu, Karlu, Timbale, Magnat and Sigma were 1.2-fold better during wintering compared to the cultivar Eugenia. In 2019, the wintering of alfalfa cultivars was similar. In 2020, the cultivars Karlu and Birutė were 1.6-fold more resistant to wintering compared to the cultivar Gea (Figure 1).

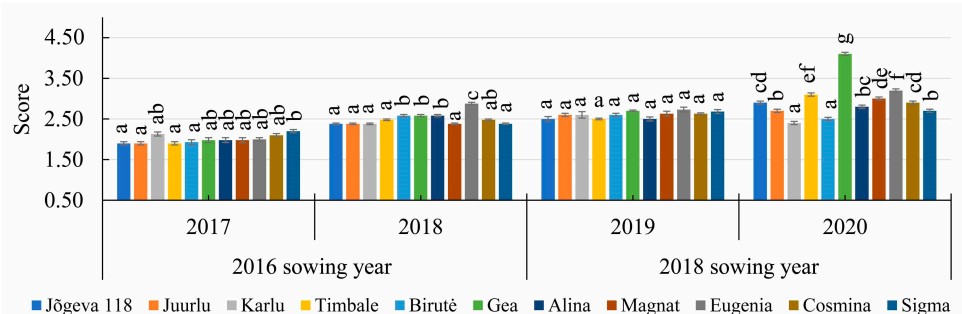

**Figure 1.** Wintering of alfalfa cultivars in seasons of 2017–2018 and 2019–2020. The differences between the cultivars with the different letters in each year are significant (*p* < 0.05). Vertical dashes indicate the mean of standard error.

In 2018, the height of alfalfa cultivars at spring regrowth was 34.4% greater compared to 2017. In 2019, the spring regrowth was 39.2% better than in 2020 (Table 2). In 2017, the height of the cultivars Timbale, Birutė, Gea, and Sigma at spring regrowth was greater compared to the cultivars Juurlu and Karlu (18.4% and 35.0%, respectively) (Figure 2). In 2018, the height at spring regrowth of the cultivars Timbale and Birutė was greater compared to the cultivars Juurlu and Karlu (45.0% and 54.7%, respectively). In 2019, the height at spring regrowth of cultivar Gea was greater compared to the cultivars Jõgeva 118, Juurlu, and Karlu (30.6%, 43.0%, and 54.0%, respectively). The cultivar Cosmina was the highest at spring regrowth in 2020 (Figure 2).

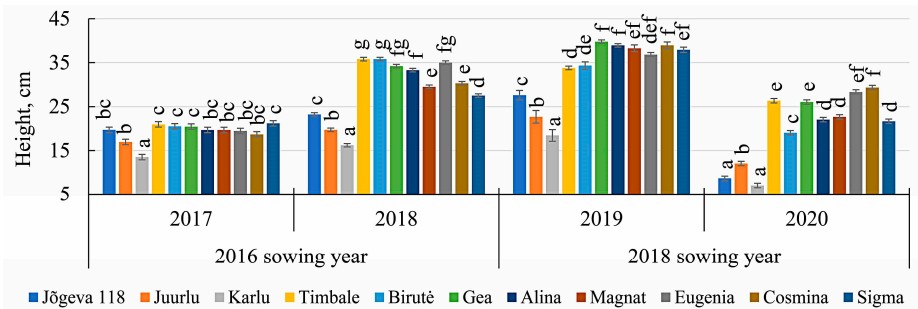

**Figure 2.** The height at the spring regrowth of alfalfa cultivars in seasons 2017–2018 and 2019–2020. The differences between the cultivars with the different letters in each year are significant (*p* < 0.05). Vertical dashes indicate the mean of standard error.

In 2017, the cultivars were significantly taller before flowering at the first, second, and third cuts compared with 2018, 17.0%, 15.7%, and 20.1%, respectively. In 2019, the height before flowering was significant greater compared to 2020, respectively at the first cut—3.7% and at the third cut—7.4% (Table 2). In 2019, the warm and very dry weather conditions influenced the height before flowering at the second cut. At the second cut, the height before flowering was 6.6% lower compared with the height before flowering at the third cut (Table 2).

In 2017, the cultivars Timbale, Birutė, Alina, and Magnat were 1.3-fold taller before flowering at the first cut compared to the cultivar Karlu. At the second cut, the cultivar Alina was 19.9% taller before flowering compared to the cultivar Jõgeva 118. At the third cut, the cultivar Gea was 1.8-fold taller before flowering compared to the cultivar Karlu.

In 2018, the height before flowering of the cultivars Alina, Eugenia, Cosmina, and Sigma was 13.9% greater at the first cut compared to the cultivar Juurlu. At the second cut, the height before flowering of the cultivars Birutė, Magnat, and Sigma was greater compared to the cultivars Jõgeva118, Juurlu, and Karlu (by 7.2%). At the third cut, the height before flowering of the cultivar Timbale was 19.5% greater compared to the cultivars Birutė and Cosmina (Figure 3a). At the first cut in 2019, the height before flowering of the cultivars Birutė and Gea was 17.1% higher compared to the cultivar Juurlu. At the second cut, the height before flowering of the cultivars Gea and Eugenia was greater, compared to the cultivars Karlu (by 47.3%). At the third cut, the height before flowering of the cultivar Eugenia was greater compared to the cultivars Timbale and Birutė (by 22.0%). In 2020, the cultivar Gea was the tallest before flowering at the first cut, compared to other cultivars. At the second cut, the height before flowering of the cultivar Timbale was the greatest and the cultivars Timbale and Gea were the tallest at the third cut (Figure 3b).

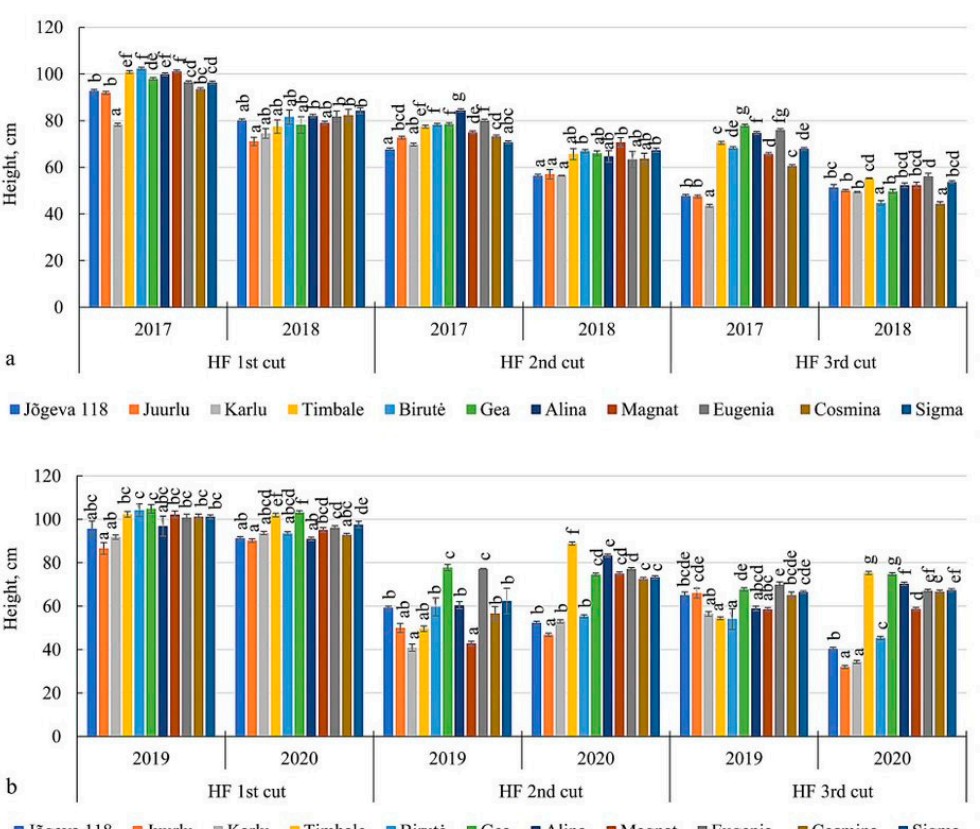

**Figure 3.** The height before flowering of alfalfa cultivars at the first, second, and third cuts in seasons 2017–2018 and 2019–2020, (**a**) 2016 sowing year; (**b**) 2018 sowing year. HF = the height before flowering. The differences between the cultivars with the different letters in each year are significant ($p < 0.05$). Vertical dashes indicate the mean of standard error.

### 3.2. The Fresh and Dry Matter Yields of Alfalfa Cultivars

The data of variance (ANOVA) showed that the fresh matter yield and the dry matter yield depended on the cultivars and year. Additionally, cultivars × year interaction was found for fresh and dry matter yields at the first, second, and third cuts in the seasons of 2017–2018 (Table 3). In seasons of 2019–2020, the significant influence of the year was determined for the fresh matter yield at the second and at the third cut and the dry matter yield at the first and at the second cuts. The cultivar × year interaction was determined for the fresh and dry matter yields at the first and second cuts (Table 3).

**Table 3.** The analysis of ANOVA and comparison of the fresh and dry matter yields of alfalfa between seasons.

| Source | df | 2016 Sowing Year (2017–2018) *p*-Value | | | | | |
|---|---|---|---|---|---|---|---|
| | | FMY 1st Cut | FMY 2nd Cut | FMY 3rd Cut | DMY 1st Cut | DMY 2nd Cut | DMY 3rd Cut |
| Cultivar | 10 | <0.0000 | <0.0000 | <0.0000 | <0.0000 | <0.0000 | <0.0003 |
| Year | 1 | <0.0000 | <0.0000 | <0.0000 | <0.0000 | <0.0000 | <0.6518 |
| Cultivar × Year | 10 | <0.0000 | <0.0000 | <0.0000 | <0.0000 | <0.0000 | <0.0000 |
| | | 2018 sowing year (2019–2020) *p*-value | | | | | |
| Cultivar | 10 | <0.0000 | <0.0000 | <0.0000 | <0.0000 | <0.0000 | <0.0000 |
| Year | 1 | <0.1465 | <0.0000 | <0.0000 | <0.0000 | <0.3667 | <0.0000 |
| Cultivar × Year | 10 | <0.0000 | <0.0000 | <0.0064 | <0.0000 | <0.0000 | <0.0027 |

| Trait | | | 2016 sowing year | | 2018 sowing year | |
|---|---|---|---|---|---|---|
| | | | 2017 | 2018 | 2019 | 2020 |
| FMY 1st cut, t ha$^{-1}$ | | | 33.3 b | 24.3 a | 29.0 a | 29.8 a |
| FMY 2nd cut, t ha$^{-1}$ | | | 17.4 a | 18.4 a | 13.2 a | 15.2 b |
| FMY 3rd cut, t ha$^{-1}$ | | | 8.5 b | 6.3 a | 12.7 b | 9.9 a |
| DMY 1st cut, t ha$^{-1}$ | | | 8.8 b | 7.0 a | 5.6 b | 4.8 a |
| DMY 2nd cut, t ha$^{-1}$ | | | 2.3 a | 4.3 b | 3.1 a | 3.1 a |
| DMY 3rd cut, t ha$^{-1}$ | | | 2.2 a | 2.2 a | 2.9 b | 2.4 a |

Note. df = degree of freedom; FMY = the fresh matter yield; DMY = the dry matter yield. Different superscripts letters "a,b" in the same row in each year indicate difference in the level of $p < 0.05$.

At the first cut in 2017, the fresh and dry matter yield depended on the very rainy weather conditions during the vegetation period. In 2017, the fresh and dry matter yields were higher at the first cut compared with 2018 (respectively, 27.0% and 20.4%). At the third cuts, the fresh matter yield was 25.9% greater compared with 2018. At the second cut in 2017, the dry matter yield was 46.5% lower compared with 2018 (Table 3). In 2020, the fresh and dry matter yield depended on the rainy weather conditions. At the first cut, the fresh matter yield was similar. At the second cut, the fresh matter yield was 13.1% greater compared with 2019. At the third cut in 2019, the fresh matter yield was 22.0% greater compared to 2020. In 2019, the dry matter yield was greater compared to 2020, respectively at the first cut—14.3%, and at the third cut—17.2% (Table 3).

In 2017, the cultivar Birutė was the most yielding based on the fresh and dry matter yields at the first cut, and was more yielding compared to the cultivar Jõgeva 118, respectively, by 57.2% and 62.8% (Figures 4a and 5a). At the second cut, the cultivars Birutė and Eugenia showed 65.1% more yield for the fresh matter yield compared to the cultivar Karlu. At the third cut, the cultivar Birutė showed a 67.5% greater yield for the fresh matter yield compared to the cultivar Karlu. In 2018, the cultivar Jõgeva 118 had a higher yield (by 57.1%) at the first cut compared to the cultivar Alina. At the second cut, the cultivar Birutė was the most yielding for the fresh and dry matter yields, and was more yielding compared to the cultivar Karlu, respectively, by 63.4% and 68.8% (Figures 4a and 5a). At the third cut, the cultivar Eugenia showed a 64.1% higher fresh matter yield compared to the cultivar Timbale (Figure 4a).

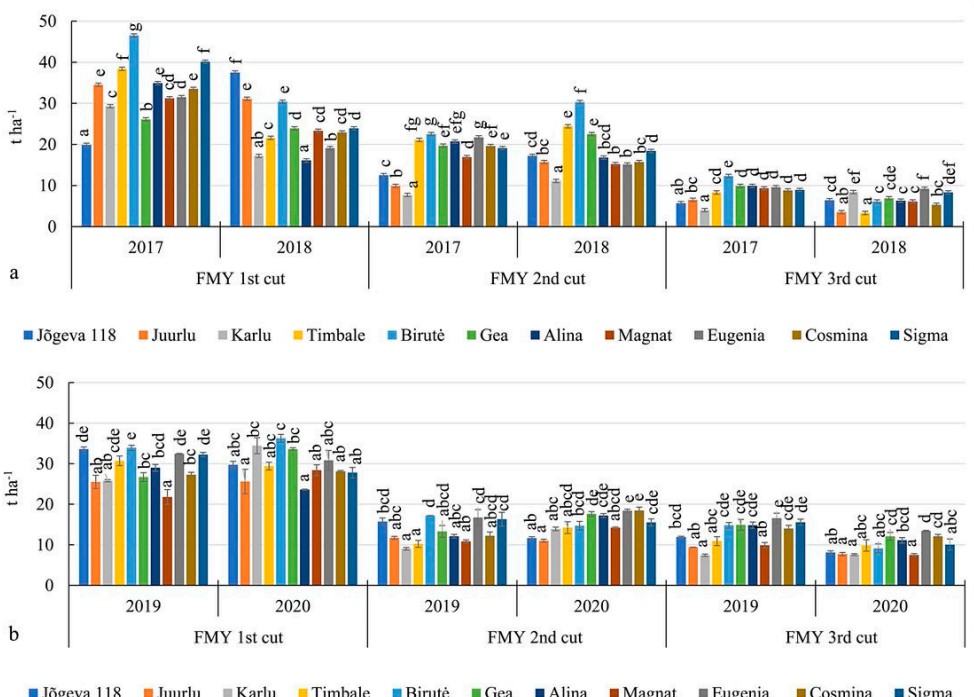

**Figure 4.** The fresh matter yield of alfalfa cultivars at the first, second, and third cuts in 2017–2018 and 2019–2020 seasons, (**a**) 2016 sowing year; (**b**) 2018 sowing year. FMY = the fresh matter yield. The differences between the cultivars with the different letters in each year are significant (*p* < 0.05). Vertical dashes indicate the mean of standard error.

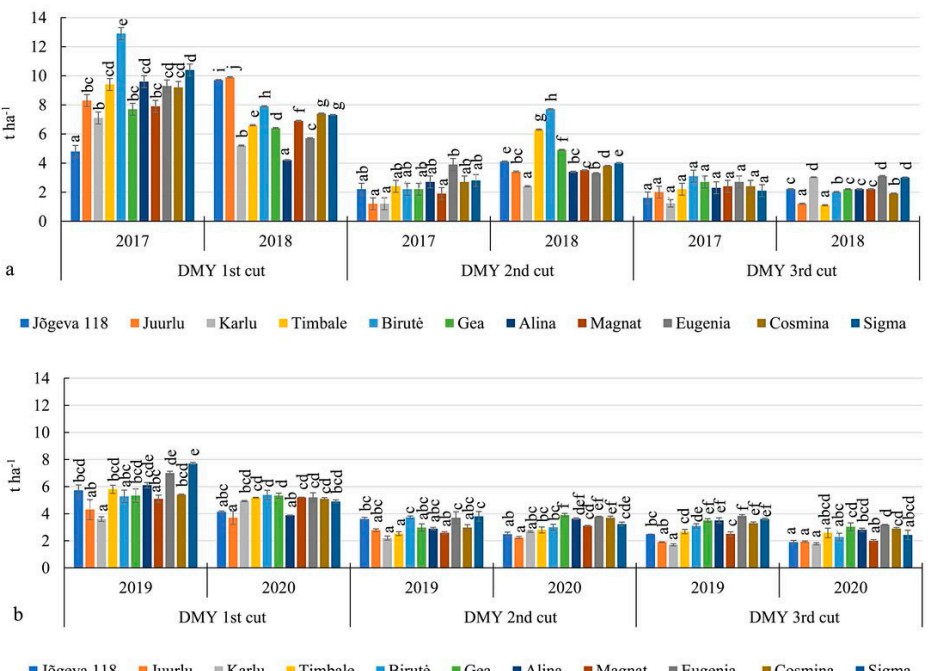

**Figure 5.** The dry matter yield of alfalfa cultivars at the first, second, and third cuts in 2017–2018 and 2019–2020 seasons, (**a**) 2016 sowing year; (**b**) 2018 sowing year. DMY = the dry matter yield. The differences between the cultivars with the different letters in each year are significant (*p* < 0.05). Vertical dashes indicate the mean of standard error.

In 2019, the cultivar Birutė was more yielding for the fresh matter yield at the first cut compared to the cultivar Magnat (by 35.7%). At the second cut, the cultivar Birutė

was more yielding compared to the cultivars Karlu and Timbale (by 44.1%). At the third cut, the cultivar Eugenia was 55.2% more yielding for the fresh matter yield compared to the cultivar Karlu (Figure 4b). In 2020, the cultivar Birutė was more yielding for the fresh matter yield at the first cut compared to the cultivars Juurlu and Alina (by 32.1%). At the second cut, the cultivars Eugenia and Cosmina were 40.2% more yielding for the fresh matter yield compared to the cultivar Juurlu. At the third cut, the cultivar Eugenia was 43.3% more yielding compared to the cultivars Juurlu, Karlu and Magnat (Figure 4b).

In 2017, the cultivar Eugenia was more yielding for the dry matter yield at the second cut compared to cultivars Juurlu, Karlu, and Magnat (by 64.1%). At the third cut, the cultivar Birutė was the most yielding for the dry matter yield. In 2018, the cultivar Juurlu was 57.6% more yielding for the dry matter yield at the first cut compared to the cultivar Alina. At the third cut, the cultivars Karlu, Eugenia, and Sigma were more yielding for the dry matter yield at the third cut compared to the cultivars Juurlu and Timbale (by 62.2%) (Figure 5a).

In 2019, the dry matter yield of the cultivar Sigma was 53.2% higher at the first cut compared to the cultivar Karlu. At the second cut, the cultivars Birutė, Eugenia, and Sigma were 36.9% more yielding compared to the cultivars Karlu and Timbale. At the third cut, the cultivar Eugenia was 55.6% more yielding for the dry matter yield compared to the cultivar Karlu. In 2020, the cultivars Birutė and Gea were 31.2% more yielding for the dry matter yield at the first cut compared to the cultivars Juurlu. At the second cut, the cultivar Gea was 42.8% more yielding for the dry matter yield compared to the cultivar Juurlu. At the third cut, the cultivar Eugenia had the highest dry matter yield (Figure 5b).

### 3.3. The Development of Spring Black Stem Leaf Spot and Downy Mildew in Alfalfa Cultivars

The result showed that development of SBSLS depended on resistance of the cultivars and year. In the seasons 2017–2018, the cultivar × year interaction was determined for SBSLS at the first, second, and third cuts (Table 4). In the seasons 2019–2020, the influence of the year was determined for SBSLS at the first, second, and third cuts. In 2017, the spring black stem leaf spot (SBSLS) developed very quickly at the first cut due to the rainy, hot weather conditions during the May–June months. The AUDPC value was higher compared with the AUDPC value at the second and at the third cuts by 3.5-fold and 2.2-fold, respectively (Table 4). At the second cut, the SBSLS developed more slowly due to the hotter and less rainy weather conditions during the July. At the third cut, the SBSLS was more developed because the weather conditions were rainy and warm during of the August–September months. At the third cut, the AUDPC value was 1.6-fold higher compared with the AUDPC value at the second cut. In 2018, SBSLS developed very slowly due to the hot–dry weather conditions during the vegetation period of alfalfa. In 2018, the AUDPC value of SBSLS at the first cut was 2.9-fold lower compared with the AUDPC value at the first cut in 2017 (Table 4).

In 2017, the cultivar Juurlu was less damaged by SBSLS at the first cut compared to the cultivar Magnat (by 1.1-fold). At the second cut, the cultivars Magnat and Eugenia were less damaged by SBSLS compared to the cultivar Birutė (by1.1-fold). At the third cut, the cultivars Eugenia and Sigma were less damaged by SBSLS compared to the cultivars Magnat and Cosmina (by 1.1-fold) (Figure 6a). At the first cut of 2018, the cultivar Alina was 1.2-fold less damaged by SBSLS compared to the cultivars Karlu, Gea, and Sigma. At the second cut, the cultivars Jõgeva118 and Karlu were 1.1-fold less damaged by SBSLS compared to the cultivars Birutė and Eugenia. At the third cut, the cultivar Jõgeva118 was 1.2-fold less damaged by SBSLS compared to the cultivar Eugenia (Figure 6a).

**Table 4.** The analysis of ANOVA and comparison of AUDPC value of SBSLS and DM between seasons.

| Source | df | 2016 Sowing Year (2017–2018) *p*-Value | | | | | |
|---|---|---|---|---|---|---|---|
| | | SBSLS 1st Cut | SBSLS 2nd Cut | SBSLS 3rd Cut | DM 1st Cut | DM 2nd Cut | DM 3rd Cut |
| Cultivar | 10 | <0.0000 | <0.0000 | <0.0000 | <0.0000 | <0.0000 | <0.0000 |
| Year | 1 | <0.0000 | <0.0000 | <0.0000 | <0.0000 | <0.0000 | <0.0000 |
| Cultivar × Year | 10 | <0.0000 | <0.0000 | <0.0000 | <0.0000 | <0.0000 | <0.0000 |
| | | **2018 sowing year (2019–2020) *p*-value** | | | | | |
| Cultivar | 10 | <0.0081 | <0.0002 | <0.1324 | <0.0000 | <0.0000 | <0.0000 |
| Year | 1 | <0.0000 | <0.0000 | <0.0000 | <0.0000 | <0.0000 | <0.0000 |
| Cultivar × Year | 10 | <0.0125 | <0.0023 | <0.0300 | <0.0000 | <0.0000 | <0.0000 |

| Trait | 2016 sowing year | | 2018 sowing year | |
|---|---|---|---|---|
| | 2017 | 2018 | 2019 | 2020 |
| SBSLS at the 1st cut, AUDPC value | 1037.9 b | 353.6 a | 254.8 a | 319.4 b |
| SBSLS at the 2nd cut, AUDPC value | 300.4 b | 184.3 a | 127.3 a | 293.5 b |
| SBSLS at the 3rd cut, AUDPC value | 476.1 b | 210.8 a | 324.5 a | 438.5 b |
| DM at the 1st cut, AUDPC value | 112.2 a | 281.2 b | 111.2 b | 0.0 a |
| DM at the 2nd cut, AUDPC value | 34.4 b | 20.5 a | 40.1 b | 0.0 a |
| DMat the 3rd cut, AUDPC value | 307.7 b | 172.5 a | 31.4 b | 0.0 a |

Note. df = degree of freedom; SBSLS = the spring black stem leaf spot; DM = the downy mildew. Different superscripts letters (a,b) in the same row in each year indicate difference in the level of *p* < 0.05.

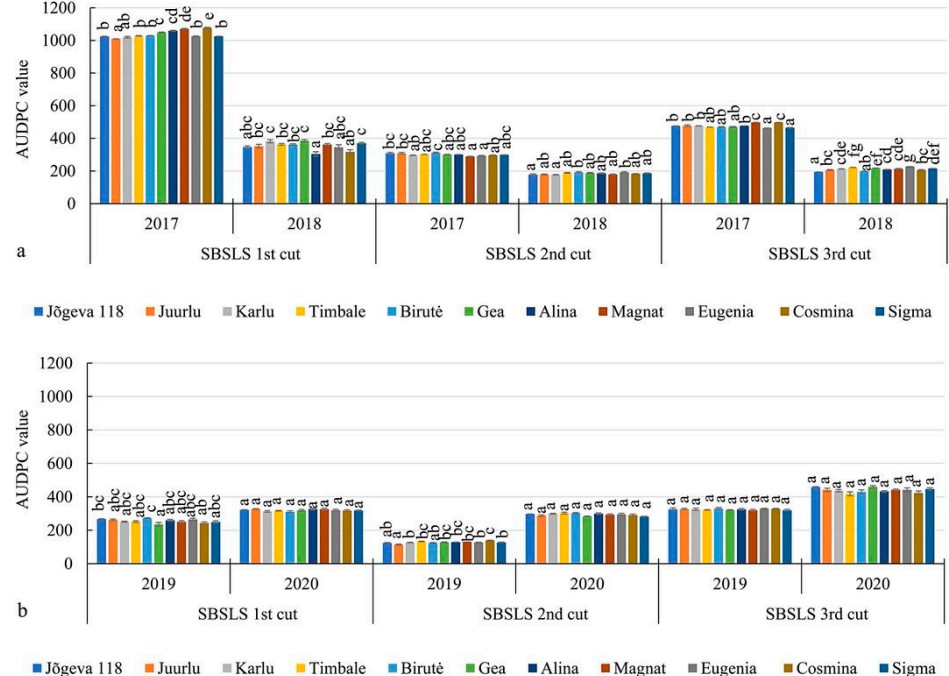

**Figure 6.** The comparison of spring black stem leaf spot in cultivars of alfalfa at the first, second, and third cuts in seasons 2017–2018 and 2019–2020, (**a**) 2016 sowing year; (**b**) 2018 sowing year. SBSLS = Spring black stem leaf spot; AUDPC = the area under the disease progress curve. The differences between the cultivars with the different letters in each year are significant (*p* < 0.05). Vertical dashes indicate the mean of standard error.

In 2019, SBSLS developed slower at the first and at the second cuts due to the very dry and hot weather conditions during May–July months. The AUDPC value of SBSLS at the first and second cuts was lower compared with the AUDPC value at the third cut (1.3-fold and 2.5-fold, respectively). In 2020, the weather conditions were very rainy and hot during the vegetation period of alfalfa cultivars, especially at the third cut. In 2020, SBSLS was

more developed at the first, second, and third cuts compared with 2019; the AUDPC value was higher by 1.3-fold, 2.3-fold, and 1.4-fold, respectively (Table 4).

In 2019, the cultivars of alfalfa differed more widely in terms of resistance to SBSLS at the first and second cuts. At the first cut, the cultivar Gea was 1.2-fold less damaged by the SBSLS compared to the cultivar Birutė. At the second cut, the cultivar Juurlu was 1.2-fold less damaged by the SBSLS compared to the cultivar Cosmina. At the third cut, the cultivars were similarly damaged by the SBSLS (Figure 6b). In 2020, the cultivars were similarly damaged by SBSLS at the first, second, and third cuts. The disease developed very rapidly during the very short vegetation period of alfalfa cultivars at each cut. The AUDPC value ranged from 310.4 to 327.3; from 281.2 to 302.2; and from 417.7 to 459.7, respectively (Figure 6b).

Significant positive correlation coefficients were observed between the fresh and dry matter yields and SBSLS at the first and second cuts, ranging from r = 0.632 * to r = 0.779 **. A strong negative correlation coefficient was observed between SBSLS at the second cut and SBSLS at the third cut (r = −0.747 **) (Table 5).

**Table 5.** Correlations relationship between agro-biological traits, 2017–2018 of 2016 sowing year.

| Traits | SR | HF1 | HF2 | HF3 | FMY2 | FMY3 | DMY1 | DMY2 | DMY3 | DM1 | DM2 | DM3 | SBSLS2 | SBSLS3 |
|---|---|---|---|---|---|---|---|---|---|---|---|---|---|---|
| W | 0.376 | 0.26 | 0.419 | 0.547 | 0.243 | 0.75 ** | −0.102 | 0.219 | 0.825 ** | 0.699 * | 0.519 | 0.425 | 0.081 | 0.118 |
| SR | | 0.917 ** | 0.846 ** | 0.849 ** | 0.912 ** | 0.631 * | 0.295 | 0.847 ** | 0.42 | 0.545 | 0.822 ** | 0.448 | 0.49 | 0.028 |
| HF1 | | | 0.782 ** | 0.747 ** | 0.822 ** | 0.654 * | 0.382 | 0.752 ** | 0.439 | 0.463 | 0.682 * | 0.378 | 0.381 | −0.013 |
| HF2 | | | | 0.859 ** | 0.726 * | 0.673 * | 0.149 | 0.552 | 0.48 | 0.668 * | 0.481 | 0.418 | 0.227 | 0.281 |
| HF3 | | | | | 0.654 * | 0.665 * | −0.028 | 0.541 | 0.53 | 0.795 ** | 0.68 * | 0.435 | 0.218 | 0.167 |
| FMY1 | | | | | 0.507 | 0.048 | 0.958 ** | 0.563 | −0.105 | −0.455 | 0.113 | −0.501 | 0.654 * | −0.525 |
| FMY2 | | | | | | 0.569 | 0.541 | 0.96 ** | 0.335 | 0.283 | 0.778 ** | 0.155 | 0.727 * | −0.237 |
| FMY3 | | | | | | | 0.165 | 0.487 | 0.953 ** | 0.725 * | 0.571 | 0.387 | 0.245 | −0.064 |
| DMY1 | | | | | | | | 0.575 | 0.029 | −0.296 | 0.173 | −0.341 | 0.632 * | −0.392 |
| DMY2 | | | | | | | | | 0.286 | 0.131 | 0.821 ** | 0.083 | 0.779 ** | −0.347 |
| DMY3 | | | | | | | | | | 0.711 * | 0.447 | 0.409 | 0.068 | −0.007 |
| DM1 | | | | | | | | | | | 0.466 | 0.642 * | −0.059 | 0.277 |
| DM2 | | | | | | | | | | | | −0.36 | | |
| DM3 | | | | | | | | | | | | | | 0.665 * |
| SBSLS2 | | | | | | | | | | | | | | −0.747 ** |

Note. W = wintering; SR = spring regrowth; H1, H2, H3 = the height before flowering at the first, second, and third cuts; FMY1, FMY2, FMY3 = the fresh matter yield at the first, second, and third cuts; DMY1, DMY2, DMY3 = the dry matter yield at the first, second, and third cuts; DM1, DM2, DM3 = downy mildew at the first, second, and third cuts; AUDPC value, SBSLS2, SBSLS3 = spring black stem leaf spot; AUDPC value, *,** = significant at $p < 0.05$ and $p < 0.01$.

In the seasons 2017–2018, the cultivar × year interaction was determined for DM at the first, second, and third cuts (Table 4). In seasons 2019–2020, the influence of the year was determined for DM at the first, second, and third cuts. The cultivar × year interaction was determined for DM at the first, second, and third cuts (Table 4). In 2017, DM developed very slowly due to the rainy weather conditions; these conditions are unfavorable for this disease. In 2017, downy mildew developed slowly at the first and at the second cuts compared with the third cut; the AUDPC value was lower 2.7-fold and 8.9-fold, respectively. In 2018, DM developed very slowly at all three cuts compared with the development of SBSLS. The AUDPC value of DM was lower compared to the AUDPC value of SBSLS at the first, second, and third cuts: 1.3-fold, 9.0-fold, and 1.2-fold, respectively. In 2019, DM developed more rapidly at the first cut, but the disease developed more slowly at the first–third cuts compared to SBSLS. The AUDPC value was lower: 2.3-fold, 3.3-fold, and 10.3-fold, respectively. In 2020, the weather conditions were unfavorable to DM development and spreads (Table 4).

In 2017, the cultivar Jõgeva118 was 1.9-fold less damaged at the first cut by DM compared to the cultivar Magnat. At the second cut, the cultivars Juurlu, Karlu, and Magnat were the least damaged by DM. At the third cut, the cultivar Eugenia was the least damaged by DM compared to cultivars Magnat and Cosmina (by 1.1-fold) (Figure 7a). In 2018, the cultivars Juurlu, Karlu, Jõgeva118, and Timbale were similarly damaged by DM at the first cut; however, the cultivar Juurlu was less damaged by DM compared to the cultivars Gea and Eugenia (by 2.7-fold). At the second cut, the cultivars of alfalfa were similarly damaged by DM. The AUDPC value of the cultivars ranged from 18.7 to 24.3. At

the third cut, the cultivars Juurlu, Karlu, and Birutė were less damaged by DM compared to the cultivar Eugenia (by 1.2-fold) (Figure 7a).

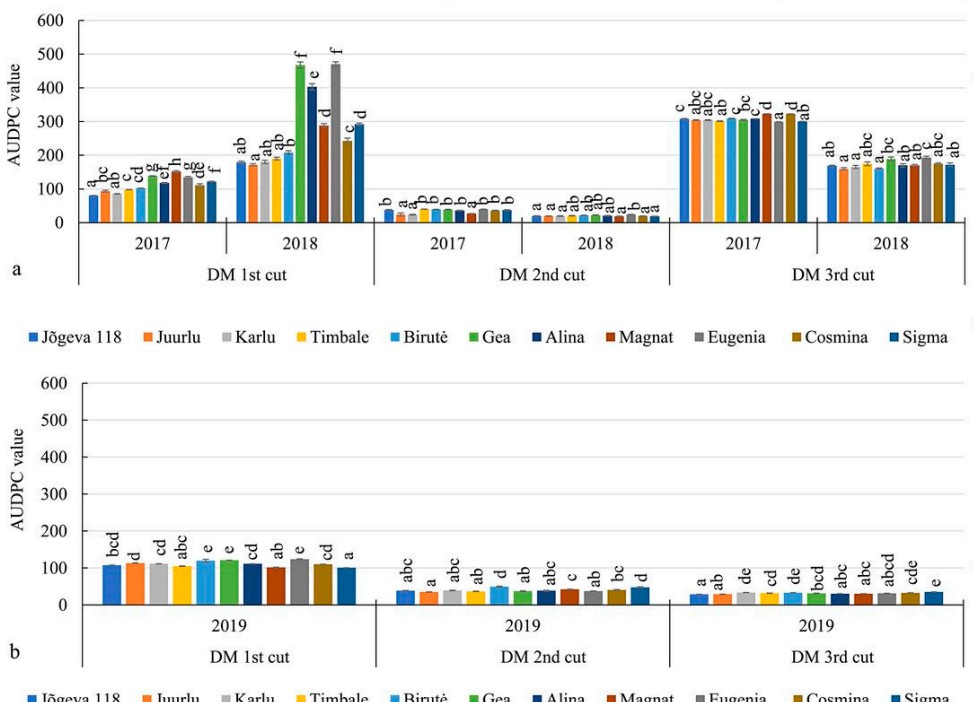

**Figure 7.** The comparison of downy mildew in cultivars of alfalfa at the first, second, and third cuts in seasons 2017–2018 and 2019–2020, (**a**) 2016 sowing year; (**b**) 2018 sowing year. DM = downy mildew; AUDPC value = the area under the disease progress curve. The differences between the cultivars with the different letters in each year are significant (*p* < 0.05). Vertical dashes indicate the mean of standard error.

In 2019, the cultivar Magnat was the least damaged by the DM at the first cut. At the second cut, the cultivar Juurlu was the least damaged by the DM and the cultivar Jõgeva 118 was the least damaged at the third cut (Figure 7b). In 2020, DM developed very slowly due to the warm, rainy weather conditions, which were unfavorable to the development of DM. It showed a correlation coefficient between DM and SBSLS at the third cut (r = 0.655 *). Additionally, the significant negative correlation coefficient was observed between SBSLS at the second cut and SBSLS at the third cut (r = −0.747 **), and between DM at the first cut and DM at the third cut (r = 0.642 *) (Table 5).

## 4. Discussion

This study has shown that the growth and development of alfalfa cultivars depended on the weather conditions, environmental factors and genetic differences between cultivars. Winter hardiness is one of the major characteristics affecting alfalfa growth and productivity in long term period [29,30]. In all experimental year, the weather conditions during wintering period were not critical for alfalfa cultivars. Additionally, in each experimental year, the wintering damages during winter period were very low, and plots of all cultivars showed very high survival rate. Cao et al. [31] suggested that the cultivars of alfalfa indigenously found in the northern regions are observed to show higher performance in winter hardiness.

The height at spring regrowth of the cultivars depended not only on the favourable weather conditions during the winter and early spring periods, but also on the genetic differences between cultivars (Figure 2). Variation in plant height among the tested genotypes could be due to genetic and environmental factors; however, environmental factors (moisture conditions, humidity, diseases occurrence, and others) influenced the growth of crops

at different stage of development [32,33]. Diriba et al. [34], Tucak et al. [24], Arab et al. [35], and Djaman et al. [36] suggested that the cultivars of alfalfa significantly differed in terms of the height of plants and determined differences between cultivars of alfalfa. Tucak et al. [37] argued that plant height was often used as a criterion when choosing superior genotypes in the early stage of selection. In our study, the height before flowering at the first, second, and third cuts depended on the genetic differences between alfalfa cultivars and the weather conditions during the winter and vegetation periods (Table 2). In 2017–2018, the strong positive correlation coefficients were determined between the height at spring regrowth and the height before flowering at the first, second, and third cut, ($r = 0.917$ **, $r = 0.846$ **, and $r = 0.849$ **), between the height at spring regrowth and the fresh matter yield at the first and third cut ($r = 0.912$ ** and $r = 0.631$ *), and between spring regrowth and dry matter yield at the second cut ($r = 0.847$ **) (Table 5). In seasons of 2019–2020, the strong positive correlation coefficients were determined between the height at spring regrowth and the height before flowering at the first, second, and third cut, ($r = 0.733$ *, $r = 0.859$ **, $r = 0.882$ **), between the height at spring regrowth and the fresh matter yield at the second and third cut ($r = 0.658$ * and $r = 0.772$ **), and between spring regrowth and the dry matter yield at the first, second, and third cut, ($r = 0.738$ **, $r = 0.703$ *, $r = 0.868$ **) (Table 6).

**Table 6.** Correlations relationship between agro-biological traits, 2019–2020 of 2018 sowing year.

| Traits | SR | HF1 | HF2 | HF3 | FMY2 | FMY3 | DMY1 | DMY2 | DMY3 | DM3 | SBSLS3 |
|---|---|---|---|---|---|---|---|---|---|---|---|
| W | 0.564 | 0.646 * | 0.681 * | 0.709 * | 0.363 | 0.496 | 0.355 | 0.427 | 0.559 | -0.18 | 0.372 |
| SR | | 0.733 * | 0.859 ** | 0.882 ** | 0.658 * | 0.772 ** | 0.738 ** | 0.703 * | 0.868 ** | 0.213 | −0.266 |
| HF1 | | | 0.705 * | 0.684 * | 0.477 | 0.51 | 0.733 * | 0.495 | 0.605 * | 0.409 | −0.145 |
| HF2 | | | | 0.945 ** | 0.707 * | 0.865 ** | 0.763 ** | 0.748 ** | 0.934 ** | 0.121 | −0.029 |
| HF3 | | | | | 0.601 | 0.785 ** | 0.733 ** | 0.668 * | 0.876 ** | 0.152 | −0.068 |
| FMY1 | | | | | 0.478 | 0.302 | 0.385 | 0.384 | 0.19 | 0.357 | 0.177 |
| FMY2 | | | | | | 0.934 ** | 0.778 ** | 0.988 ** | 0.875 ** | 0.312 | 0.214 |
| FMY3 | | | | | | | 0.743 ** | 0.94 ** | 0.978 ** | 0.216 | 0.091 |
| DMY1 | | | | | | | | 0.788 ** | 0.769 ** | 0.467 | −0.06 |
| DMY2 | | | | | | | | | 0.895 ** | 0.258 | 0.26 |
| DM2 | | | | | | | | | | 0.651 * | −0.103 |
| SBSLS2 | | | | | | | | | | | −0.647 * |

Note. W = wintering; SR = spring regrowth; H1, H2, H3 = the height before flowering at the first, second, and third cuts; FMY1, FMY2, FMY3 = the fresh matter yield at the first, second, and third cuts; DMY1, DMY2, DMY3 = the dry matter yield at the first, second, and third cuts; DM2, DM3 = downy mildew at the second and third cuts; SBSLS2, SBSLS3 = spring black stem leaf spot at the second and third cuts; *,** = significant at $p < 0.05$ and $p < 0.01$.

Abd et al. [38] and Arab et al. [35] reported significant differences between genotypes and found interaction between genotypes and year for the fresh and dry matter yields. Seiam and Mohamed [39] found highly significant differences in years and seasons for the fresh and dry matter yields. Environmental conditions and plant genetics play a significant role in the variation in the fresh matter and dry matter yields among cultivars [40–42]. Studies are based on the fact that the cutting frequency has a significant effect on forage yield and yield components in alfalfa, and the crop harvesting cycle has a significant effect on other parameters, such as height [43–45]. The harvest intervals or frequency are influenced by environmental factors, which also affect alfalfa growth characteristics [44,46]. Additionally, alfalfa productivity is affected by its fall dormancy level, which has correlation with forage yield, agronomy characteristics, and nutritive value [36]. In our research, we used the three cuttings over the year; the fresh and the dry matter yields and the height before flowering were significantly lower at the second and third cuts. The plant height is strongly associated with the total fresh and dry crop yields [37,47,48]. This fact is supported by the strong positive correlation coefficients between the height before flowering at the first, second and third cuts and the fresh, dry matter yields at the first, second and third cuts, ranging from $r = 0.54$ * to $0.822$ ** in 2017–2018, and ranging, from $r = 0.605$ * to $r = 0.876$ ** in 2019–2020 (Tables 5 and 6).

In previous research on the period 2014–2017, the total fresh matter and total dry matter yields were similar in all experimental years and ranged, from 46.3 t ha$^{-1}$ to 53.7 t ha$^{-1}$, and from 11.0 t ha$^{-1}$ to 13.0 t ha$^{-1}$ [49]. In this study, during the seasons 2017–2020, the total fresh matter yield ranged from 49.0 t ha$^{-1}$ to 59.1 t ha$^{-1}$, and the total dry matter yield ranged from 10.3 t ha$^{-1}$ to 13.5 t ha$^{-1}$. Veronesi et al. [40], Lee et al. [50], and Hatfield et al. [51] suggested that alfalfa forage yield depends, to a great extent, on various climatic factors, with precipitation and temperature and cutting number. Wang et al. (2021) [52] argued that the forage yield depended on the environmental conditions under six harvests, and these authors found the total forage yield ranged from 24.4 t ha$^{-1}$ to 32.7 t ha$^{-1}$. Results clearly demonstrated that the fresh and dry matter yield depended on the cutting time, precipitation amounts, and temperature in each year. Additionally, cutting alfalfa three times within a year, the cultivars revealed the potential of the total harvest. Svirskis [53] argued that the dry matter yield of foreign cultivars was 30% lower compared to Lithuanian cultivars. Annicchiarico, [54] and Monirifar, [47] found a strong positive correlation coefficient between the fresh and dry yields. Additionally, Diriba et al. [34] argued that the differences in value of the dry matter yield might be observed due to the attributed cultivar or environmental and their interaction. The fresh and dry matter yields depended on the weather conditions, which varied not only by experimental year, but, also between at the first, second and third cuts. In the periods 2017–2018 and 2019–2020 very strong positive correlation was found between the fresh and dry matter yields at the first, second, and third cuts, ranging from r = 0.743 ** to r = 0.988 ** (Tables 5 and 6).

SBSLS causal agent overwinters in abundancy in plants residues and disease causal pathogen spreads quickly in early spring. However, disease damages essential plant surface only after several tens of days due to slow pathogen development in plant tissues [55]. Disease development can be delayed by insufficient precipitations. In our study, the development of the disease varied due to the different amount of precipitation at the first, second, and third cuts. Djebali [20] argued that SBSLS causes yield and forage quality losses in alfalfa crops. The development of SBSLS depended on the rainy and hot weather conditions; however, the disease development and spread was limited due to the very short period of vegetation at each cut. The cultivars were less damaged by disease in each cut, and the losses of the dry and fresh matter yields were very low. This was shown by the significant positive correlation coefficients between the fresh and dry matter yields and SBSLS at the first and second cuts (Table 5).

Previous research showed that SBSLS developed and spread more rapidly in the seed crops of alfalfa, not only due to the rainy and very hot weather conditions, but also due to longer period of the disease development during the overall vegetation period of alfalfa. In 2010, the AUDPC value of SBSLS between genotypes of alfalfa varied from 2263 to 2928 [56]. In 2010, the cultivars Magnat, Alina, and Sigma were less damaged by SBSLS compared to the cultivar Birutė; the AUDPC values of these cultivars were 1.1-fold lower. However, the cultivar Birutė was less damaged by SBSLS compared to the cultivar Cosmina the AUDPC value was 1.0-fold lower. In 2009, the cultivars Magnat, Alina, Cosmina, and Sigma were less damaged by SBSLS compared to the cultivar Birutė. The AUDPC value of the cultivars was lower by 1.8-fold, 1.4-fold, 2.4-fold, and 1.5-fold, respectively. In 2011, the AUDPC value of the cultivar Magnat was lower by 1.2-fold, and the AUDPC of the cultivars Alina, Cosmina, and Sigma was lower by 1.1-fold compared to the cultivar Birutė [57]. Ellwood et al. [58] and Kamphuis et al. [59] suggested that alfalfa, as a cross-pollinating plant, consists of individuals which differ by resistance. Resistance to disease depends on polygenes. Development of a new population with greater resistance to SBSLS should be carried out under greenhouse or laboratory conditions that sustainably differentiate cultivars and plants in terms of resistance [55,60,61]. Screening populations' resistance to SBSLS at the seedling stage can denote the most resistant seedlings [20]. Additionally, in this research, we evaluated the more resistant cultivars of alfalfa, which were the least damaged by SBSLS at the first, second, and third cuts.

Downy mildew (DM) is an important disease in crops of alfalfa, which reduces yields and crop quality [17]. The disease spreads when the weather conditions are cold and humid, but not rainy [22]. The result showed that development of DM depended on the resistance of the cultivars and the year.

Previous research showed that DM developed and spread very slowly in the crops of alfalfa, due to the hot, dry weather conditions. In 2011, the disease developed during the vegetation period of alfalfa and the cultivars differed in terms of resistance to DM. The AUDPC value ranged from 40 to 266. In 2009, the disease spread more rapidly due to the wet and cool weather conditions and the cultivars differed in terms of their resistance to disease. The AUDPC value ranged from 140 to 1838 [62]. The cultivar Birutė was less damaged by DM compared to the cultivars Magnat, Alina, Cosmina, and Sigma. In 2011, the AUDPC value of the cultivar Birutė was lower compared to the cultivars Magnat, Alina, Cosmina, and Sigma, by 1.6-fold, 1.7-fold, 1.8-fold, and 2.0-fold, respectively. In 2009, the AUDPC value of the cultivar Birutė was lower compared to the cultivars Magnat, Alina, Cosmina, and Sigma by 4.3-fold, 6.4-fold, 5.4-fold, and 4.9-fold, respectively. Additionally, in 2010, the AUDPC value of the cultivar Birutė was lower compared to the cultivars Magnat, Alina, Cosmina, and Sigma by 2.4-fold, 3.8-fold, 2.6-fold, and 2.9-fold, respectively [57]. Yaege, and Stutevile [63] and Nagl et al. [64] argued that resistance to downy mildew depends on combinations of mono and polygenes. The alfalfa genotypes consist of plants which differ in terms of resistance. In our research, we did not find any cultivars completely resistant to disease. However, the most resistant cultivars, which were the least damaged by DM, had the lowest AUDPC value.

## 5. Conclusions

The study showed that the cultivars of alfalfa differed in terms of agro-biological traits under contrasting seasons of 2017–2018 and 2019–2020. The weather conditions during the winter period were favorable for wintering of alfalfa cultivars in all experimental years. Additionally, the height of alfalfa cultivars during the spring regrowth and before flowering differed in each cut and depended on the weather conditions. SBSLS spread more rapidly compared with DM in all experimental years. The fresh and dry matter yields at the first, second, and third cuts varied year by year. The cultivar Birutė from Lithuania was distinguished by wintering, the plant height at spring regrowth, the height before flowering, and the fresh and dry matter yields. The cultivar Timbale from France was distinguished by wintering, spring regrowth, and the height before flowering. The cultivars Alina, Magnat, and Sigma from Romania were distinguished by the height before flowering; however, the cultivar Magnat was less damaged by downy mildew. The cultivars Jõgeva 118, Juurlu, and Karlu were distinguished by wintering and were less damaged by diseases. The cultivars from Italy were distinguished by the fresh and dry matter yields and the height before flowering (cultivar Eugenia and cultivar Gea). These results highlight the most important agro-biological traits of the studied cultivars, which will enable breeders to effectively select genotypes for further cultivars development.

**Author Contributions:** A.L. conceived and designed the experiment, performed the experiments, analyzed the data, prepared figures, and tables, and wrote and reviewed drafts of the paper; V.D. and Ž.L. performed statistical analysis and prepared figures and tables; A.L., V.D., Ž.L. and E.N. reviewed drafts of the paper and approved the final draft. All authors have read and agreed to the published version of the manuscript.

**Funding:** This research received no external funding.

**Informed Consent Statement:** Not applicable.

**Data Availability Statement:** Not applicable.

**Acknowledgments:** The paper presents research findings obtained through the long-term research programme 'Genetics, biotechnology and breeding for plant biodiversity and innovative technologies' implemented by Lithuanian Research Centre for Agriculture and Forestry.

**Conflicts of Interest:** The authors declare no conflict of interest.

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
