# Peer review of "Diversity of Agro-Biological Traits and Development of Diseases in Alfalfa Cultivars during the Contrasting Vegetation Seasons"

_sustainability, doi:10.3390/su15021445_

Round 1

Reviewer 1 Report

Current study  gives interesting and valuable information about different cultivars for breeding purpose.  Suggestions and comments  are listed in the manuscript.

Author´s  It is difficult to understand the Results when differences between cultivars are presented in fold´s. It is difficult to follow the figures, the ‘fold‘ does not become evident (perhaps percentage is better). 

Extensive English language editing is required.

Reviewer 2 Report

The authors evaluated the agro-biological traits and leave spot and downy mildew disease development in eleven alfalfa cultivars of different origin. I consider the work fairly conceived and methodology and resulting data are consistent with the overall objective of the study. Although this manuscript presents important data to the scientific community, it yet has to be revised for the following minor issues before it can be considered for publication.

1) What is the novelty of present study and how it comes unique from the previous studies? 

2) Results are too long with redundant interpretation. Make them concise.

3) Discussion can further be improved by adding more relevant literature and citing some recent references. For example, following recent studies could be added to compare and support your results with relation to previous research.

Wang et al. 2021 (Dynamics of Spring Regrowth and Comparative Production Performance of 50 Autumn-Sown Alfalfa Cultivars in the Coastal Saline Soil of North China)

Wang et al. 2022 (Evaluation of Different Shallow Groundwater Tables and Alfalfa Cultivars for Forage Yield and Nutritional Value in Coastal Saline Soil of North China)

Reviewer 3 Report

Selecting different alfalfa varieties from different sources to evaluate agronomic traits and diseases is conducive to the selection of local alfalfa varieties in the production process. In this paper, 11 varieties from different sources were collected and evaluated, their plant height, biomass, downy mildew and other indicators were identified, and the differential performance of different varieties was obtained. This is of great significance for recommending suitable varieties for Lithuania, and provides reference data for other countries to understand the performance of these varieties in Lithuania.

Some details in the article still need to be corrected by the author.

Line 51  "de Bary.," is there an extra "."

Line 109  The formula should be explained in more detail, but the present version seems difficult to understand.

Line 105 SBSLS should explain its specific meaning, which does not seem very clear here

The figure quality is not very good in the full text

Reviewer 4 Report

The work is experimental work that brings little to scientific development. analyzes the behavior of different cultivars. It's a job better suited for a genetic improvement journal. I cannot visualize what this essay brings in terms of methodological knowledge to the journal community. Or the authors most to show the relation between the journal subject and the article

Reviewer 5 Report

I recommend the publication of the article because evaluating the agrobiological characteristics and selecting the most promising cultivars of alfalfa in different contrasting growing seasons is interesting from a botanical point of view, because in the future these studies will be important with regard to climate change.

The aim and objectives of the article have been stated and are very fascinating.

The evaluation of new botanical species of alfalfa is an important topic especially in organic farming with regard to the reduction of synthetic products and chemical fertilisers and resistance to biotic and abiotic stresses.  The work is certainly of international interest and the format applied is certainly suitable for a research paper. The work is original, of particular interest and can certainly stimulate research on this topic. The length of the article is good for the journal and the graphs and tables are clear and easy to understand. The conclusion summarises the aims of the work and future prospects.

Round 2

Reviewer 1 Report

At this point the manuscript has been significantly improved. Thank You!

Reviewer 4 Report

I consider that are improvements but its a very tevhunical article
